

# Chaotic oceanic excitation of low-frequency polar motion variability

Lara Börger[1], Michael Schindelegger[1], Mengnan Zhao[2], Rui M. Ponte[2], Anno Löcher[1], Bernd Uebbing[1], Jean-Marc Molines[3], and Thierry Penduff[3]

[1]Institute of Geodesy and Geoinformation, University of Bonn, Bonn, Germany
[2]Atmospheric and Environmental Research (AER), Inc., Lexington, MA, USA
[3]Université Grenoble Alpes, CNRS, INRAE, IRD, Grenoble INP, Institut des Géosciences de l'Environnement (IGE), Grenoble, France

**Correspondence:** Lara Börger (lboerger@igg.uni-bonn.de)

**Abstract.** Studies of Earth rotation variations generally assume that changes in non-tidal oceanic angular momentum (OAM) manifest the ocean's direct response to atmospheric forces. However, fluctuations in OAM may also arise from chaotic intrinsic ocean processes that originate in local nonlinear (e.g., mesoscale) dynamics and can map into motions and mass variations at basin scales. To examine whether such random mass redistributions effectively excite polar motion, we compute monthly OAM anomalies from a 50-member ensemble of eddy-permitting global ocean/sea-ice simulations that sample intrinsic variability through a perturbation approach on model initial conditions. The resulting OAM (i.e., excitation) functions, $\hat{\chi}^O$, are examined for their spread, spectral content, and role in the polar motion excitation budget from 1995 to 2015. We find that intrinsic $\hat{\chi}^O$ signals are comparable in magnitude to the forced component at all resolved periods except the seasonal band, amounting to ∼46% of the total oceanic excitation (in terms of standard deviation) on interannual time scales. More than half of the variance in the intrinsic mass term contribution to $\hat{\chi}^O$ is associated with a single, global mode of random bottom pressure variability, likely generated by nonlinear dynamics in the Drake Passage. Comparisons of observed interannual polar motion excitation against the sum of known surficial mass redistribution effects are sensitive to the representation of intrinsic $\hat{\chi}^O$ signals: Reductions in the observed excitation variance can be as high as 68%, or as low as 50% depending on the choice of the ensemble member. Chaotic oceanic excitation thus emerges as a new factor to consider when interpreting low-frequency polar motion changes in terms of core-mantle interactions or employing forward-modeled OAM estimates for Earth rotation predictions.

## 1 Introduction

Changes in ocean currents and the global distribution of ocean mass imply variability in oceanic angular momentum (OAM). Transfer of portions of this momentum to the solid Earth causes measurable fluctuations in the latter's rotation, comprising both changes in angular velocity and the movement of a conventional pole relative to Earth's crust–commonly referred to as wobble, or polar motion (Gross, 1992). Past studies (e.g., Ponte et al., 1998; Gross et al., 2003; Bizouard and Seoane, 2010; Harker et al., 2021; Afroosa et al., 2021; Börger et al., 2023) have shown that variability in non-tidal OAM is indeed a prominent source for wobble excitation on time scales from days to several years, modulating or even exceeding the rotational contributions of





other geophysical fluids. An assumption in all of these works is that the path to accurate OAM estimates is to encode known physics in a numerical ocean model and integrate that very model in time under the necessary external (mostly atmospheric) forcings, either with or without data constraints. However, such an approach overlooks the possibility of a significant random component in the evolving ocean state, emerging internally within the fluid irrespective of atmospheric fluctuations. Evidently, any such intrinsic, chaotic variability would be inaccessible—or at least challenging to capture—with usual means of forward or inverse modeling.

A well-known example of the ocean's intrinsic variability is the nonlinear, turbulent flow as manifested in mesoscale eddies. While these eddies operate at typical scales of O(10–100) km and O(10–100) days (Rhines, 2019), their kinetic energy is known to cascade toward much larger spatial and longer temporal scales (e.g., Venaille et al., 2011; Arbic et al., 2014; Sérazin et al., 2018). In detail, ocean simulations performed in the eddying regime (at grid spacings $\leq 1/4°$) have demonstrated that interactions between relatively short-lived mesoscale eddies and large-scale baroclinic instability can generate chaotic interannual to secular variability in regional sea level (e.g., Llovel et al., 2018; Carret et al., 2021), large-scale volume transports (Leroux et al., 2018; Cravatte et al., 2021), or the eddy field itself (Hogg et al., 2022). Moreover, the same simulations suggest that nonlinear dynamics feed random fluctuations in ocean mass between entire basins (Zhao et al., 2023). Given this large-scale chaotic imprint on dynamical variables, intriguing questions arise as to whether oceanic intrinsic variability also affects OAM and what such random variability potentially means for the interpretation of polar motion observations.

We address these open questions using output from a $1/4°$ large-ensemble ocean hindcast experiment, as analyzed in numerous previous studies of oceanic chaos and its inverse cascades (e.g., Leroux et al., 2018; Sérazin et al., 2018; Cravatte et al., 2021; Hogg et al., 2022). Focus is lain on the period from 1995 to 2015 and particularly the interannual band, where oceanic effects are a leading cause for wobble excitation, along with continental hydrology (Gross et al., 2003; Wińska, 2022; Börger et al., 2023). We separate the ensemble OAM estimates into forced (i.e., atmospheric-driven) and intrinsic components and assess their contributions to the observed wobble excitation in the presence of mass change and motion elsewhere in the Earth system. This in turn calls for a consideration of all known climatic excitations of polar motion, as described below.

## 2 Methods

### 2.1 Basic framework and data

Polar motion is a two-dimensional quantity, conventionally expressed as complex-valued angular distance $\hat{p} = \hat{p}(t) = p_1 + \mathrm{i}p_2$, where $t$ indicates time and subscripts 1 and 2 refer to longitudes of $0°$ and $90°$E in the terrestrial reference frame. Under the conservation of angular momentum (Munk and MacDonald, 1960; Moritz and Mueller, 1987), the equation governing the excitation of observed variations in $\hat{p}$ is (e.g., Brzeziński, 1992)

$$\hat{p} + \mathrm{i}\hat{\sigma}_{\mathrm{c}}^{-1}\dot{\hat{p}} = \hat{\chi} \,, \tag{1}$$

where $\hat{\chi} = \chi_1 + \mathrm{i}\chi_2$ is the polar motion excitation function (Barnes et al., 1983; Gross, 2007), the dot denotes the time derivative, and $\hat{\sigma}_{\mathrm{c}} = 2\pi(1 + \mathrm{i}/2Q_{\mathrm{c}})/T_{\mathrm{c}}$ is the complex-valued Chandler frequency with standard values for period $T_{\mathrm{c}} = 433.0$ days and





quality factor $Q_c = 179$ (Gross et al., 2003). Given proper observations of $\hat{p}$, the deconvolution on the left side of Eq. (1) yields the "geodetic" (or observed) excitation of polar motion, labeled as $\hat{\chi}^*$ throughout this work. We specifically use rotation data from the daily-sampled SPACE2018 series (Ratcliff and Gross, 2019) that employs a Kalman filter to adjust measurements from the primary space-geodetic techniques. We remove long-period tidal effects in polar motion using the conventional model

of Petit and Luzum (2010), deconvolve $\hat{p}$, and finally average the daily $\hat{\chi}^*$ values to monthly estimates.

The practical problem defined by Eq. (1) lies in identifying the causes for the observed excitation from angular momentum changes in geophysical fluids. As is customary (e.g., Wahr, 1982), we consider excitations both due to the redistribution of mass and particle movement, that is,

$$\hat{\chi} \quad = \quad \hat{\chi}^{\mathrm{m}} + \hat{\chi}^{\mathrm{v}} , \tag{2}$$

where the mass or pressure term (superscript m) accounts for perturbations in inertia products, and the motion term (superscript v for velocity) represents the fluid's relative angular momentum. Formulae for computing $\hat{\chi}^{\mathrm{m,v}}$ as area integrals over globally distributed mass changes and horizontal velocities are given in Appendix A. Note that these geophysical excitation functions are also called "effective angular momentum functions" (Barnes et al., 1983; Dickman, 2003), where the term "effective" pertains to the non-rigid component of the Earth's response to the excitation process. Below we occasionally refer to the oceanic $\hat{\chi}$

estimate as OAM function, thus omitting the prefix "effective" for convenience.

In comparisons with $\hat{\chi}^*$, we consider excitations from four main climate system components, comprising the atmosphere (denoted to as $\hat{\chi}^{\mathrm{A}}$), the ocean ($\hat{\chi}^{\mathrm{O}}$), the continental hydrosphere ($\hat{\chi}^{\mathrm{H}}$), and the cryosphere as represented by the Greenland and Antarctica ice sheets ($\hat{\chi}^{\mathrm{C}}$). The satellite data underlying $\hat{\chi}^{\mathrm{H}}$ and $\hat{\chi}^{\mathrm{C}}$, along with the ensemble approach to oceanic excitation (Sect. 2.2) constrains us to have monthly sampling. For consistency with the ocean model output analyzed here, we use atmo-

spheric angular momentum time series from the ERA-Interim reanalysis of the European Centre for Medium-Range Weather Forecasts (Dee et al., 2011). The $\hat{\chi}^{\mathrm{A}}$ series is an extension of a previous calculation by us (Schindelegger et al., 2013) and accounts for excitation signals associated with the ocean's inverted barometer (IB) response to atmospheric pressure loading. Figure 1 shows that on interannual time scales (here periods $\geq 14$ months[1]), the atmosphere is somewhat less effective than the ocean in exciting Earth's wobbles, particularly through $\chi_1$ (cf. Gross et al., 2003). Peak-to-peak amplitudes in $\chi_2^{\mathrm{A}}$ occasionally

attain 15 mas (milliarcseconds) and are largely governed by mass redistribution over Northern hemisphere landmasses (Nastula and Salstein, 1999).

Contributions to $\hat{\chi}^{\mathrm{m}}$ from continental hydrology—or equivalently, terrestrial water storage (TWS)—and the cryosphere are deduced from time-variable gravity fields, as practiced in previous works (e.g., Chen et al., 2013; Adhikari and Ivins, 2016; Nastula et al., 2019; Göttl et al., 2021). To cover the same 21-year period (1995–2015) as for our oceanic excitation series, we

use a long-term reconstruction of monthly gravity fields based on satellite tracking data (Löcher and Kusche, 2021, updated version). The approach pursued for this reconstruction is to expand surface mass changes from the Gravity Recovery and Climate Experiment/-Follow On (GRACE/-FO, Tapley et al., 2019) mission into empirical orthogonal functions (EOFs) and

---

[1]This cutoff period was chosen to effectively suppress the seasonal components in the excitation time series. For studying the excitation of the Chandler wobble at $T_c = 433.0$ days, which is not attempted here, a different approach to analysis would be required (see Gross, 2000).





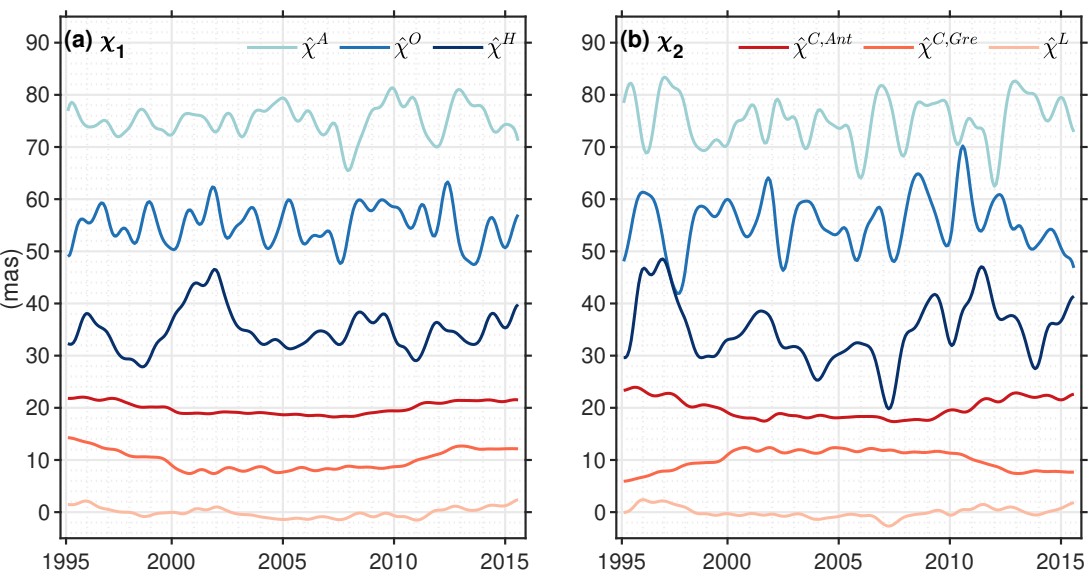

**Figure 1.** Modeled excitation $\hat{\chi} = \chi_1 + i\chi_2$ of interannual polar motion (in milliarceseconds), from 1995 through 2015, separated into contributions from atmosphere ($\hat{\chi}^A$, light blue), ocean ($\hat{\chi}^O$, blue), terrestrial hydrology ($\hat{\chi}^H$, dark blue), the Antarctic Ice Sheet ($\hat{\chi}^{C,Ant}$, red), the Greenland Ice Sheet ($\hat{\chi}^{C,Gre}$, orange), and gravitational attraction and loading ($\hat{\chi}^L$, creme). Interannual signals were obtained by removing each time series' trend and seasonality and low-pass filtering the residual series with a cutoff period of 14 months. Four months were clipped both at the beginning and at the end of the time series to avoid filter artifacts. The $\hat{\chi}^O$ estimate is from an arbitrary OCCIPUT ensemble member.

use the leading EOFs as basis functions for gravity field modeling in the processing of the tracking observations. The solution has no temporal gaps and preserves the spatial resolution of GRACE/-FO gravity fields (i.e., wavelengths of ∼300 km) also

for times prior to GRACE. We synthesize the monthly varying Stokes coefficients as equivalent water heights across Earth's surface, mask out the ocean, and partition the mass anomalies into changes over the Greenland Ice Sheet, the Antarctic Ice Sheet, and land. Spatial leakage effects near continental boundaries and ice sheet margins are mitigated using an iterative forward modeling algorithm (Chen et al., 2015); see Appendix B for details. Figure 1 illustrates that hydrological mass changes drive large (∼10–15 mas) interannual to intradecadal variations in polar motion excitation (Adhikari and Ivins, 2016; Meyrath

and Van Dam, 2016). De-trended time series for cryospheric sources ($\hat{\chi}^C$) reveal residual long-term drifts that bear on the accelerated melting of polar ice sheets (Chen et al., 2013; Deng et al., 2021).

In addition to $\hat{\chi}^{A,O,H,C}$, we account for non-uniform mass changes in the ocean due to gravitational attraction and loading (GAL), defining the excitation function $\hat{\chi}^L$. Processes described by GAL are the continent-ocean net mass transfer, the gravitational attraction of water toward mass anomalies anywhere in the Earth system, the crustal deformation under these loads,

and the ensuing changes in Earth's gravitational field (e.g., Clarke et al., 2005). The surface mass variations arising from these effects are long-wavelength in nature, leading to small but non-negligible angular momentum changes (Quinn et al., 2015; Adhikari and Ivins, 2016, Fig. 1). We calculate the GAL effects on ocean mass and distribution by solving the sea level equation





(Farrell and Clark, 1976), assuming elastic (i.e., instantaneous) deformation of a rotating Earth and a purely static adjustment in the ocean. The considered loads are gridded monthly anomalies of TWS, cryospheric mass, and IB-corrected atmospheric

pressure, taken from the same sources as those adopted for our $\hat{\chi}^{\mathrm{A,H,C}}$ estimates; see Appendix C.

## 2.2  Oceanic excitation

We study the externally forced and chaotic intrinsic components of oceanic excitation based on a 50-member ensemble of eddy-permitting ocean/sea ice simulations from the OceaniC Chaos–ImPacts, strUcture, predicTability (OCCIPUT) project (Penduff et al., 2014). The OCCIPUT large ensemble was generated using a global $1/4°$ tri-polar grid setup of the Nucleus for

European Modelling of the Ocean engine for Boussinesq ocean dynamics and thermodynamics. The full ensemble hindcast spans 56 years (1960–2015). Following a common 21-year spin-up simulation, a small stochastic perturbation was applied in the density equation of each member throughout the year 1960, triggering the growth and subsequent saturation of the ensemble spread (Penduff et al., 2014; Brankart et al., 2015; Bessières et al., 2017; Leroux et al., 2018). These 50 realizations of the time-evolving ocean state were then integrated forward from their respective, slightly perturbed initial conditions and

driven by the same 6-hourly atmospheric data from the DRAKKAR forcing set DFS 5.2 (Dussin et al., 2016), which itself is based on ERA-Interim.

To keep the involved data volumes manageable, we analyze the OCCIPUT ensemble output from 1995 to 2015 (cf. Zhao et al., 2021) and focus on monthly mean values. Specifically, we evaluate the mass and motion terms of $\hat{\chi}^{\mathrm{O}}$ (see Appendix A) from monthly averages of ocean bottom pressure, $p_{\mathrm{b}} = p_{\mathrm{b}}(\mathbf{x}, t)$, and vertically-averaged northward and eastward currents,

$\overline{u}(\mathbf{x}, t)$ and $\overline{v}(\mathbf{x}, t)$, where $\mathbf{x}$ represents horizontal space. Each of the three variables $\xi(\mathbf{x}, t) \in (p_{\mathrm{b}}, \overline{u}, \overline{v})$ consists of externally forced and intrinsic signals, i.e.,

$$\xi(\mathbf{x}, t, \mu) = \xi^{\mathrm{f}}(\mathbf{x}, t) + \xi^{\mathrm{i}}(\mathbf{x}, t, \mu) \; , \tag{3}$$

where $\mu$ is a running index over the 50 ensemble members, superscript f indicates the forced variability common to all members, and i identifies the intrinsic signal. As in previous studies (see in particularly Hogg et al., 2022), we estimate the forced

component at a given month and location by the ensemble mean $\xi^{\mathrm{f}}(\mathbf{x}, t) = \langle \xi(\mathbf{x}, t, \mu) \rangle$, adopting $\langle \cdot \rangle$ as the ensemble-mean operator. Upon separating signals as per Eq. (3), we compute angular momentum time series (Appendix A) for forced and intrinsic contributions to $(p_{\mathrm{b}}, \overline{u}, \overline{v})$, as well as for their sum. Hence, intrinsic variability in oceanic excitation (abbreviated as $\hat{\chi}^{\mathrm{O,i}}$) is distinguished from forced variability ($\hat{\chi}^{\mathrm{O,f}}$) at the level of spatially distributed dynamical variables, and not based on excitation time series.

Note that throughout this work, $p_{\mathrm{b}}$ denotes ocean bottom pressure anomalies, obtained by removing for each ensemble member a common time-mean reference pressure field and a spatial mean $p_{\mathrm{b}}$ value per month and ensemble member. The latter correction prevents small biases or drifts in $\hat{\chi}^{\mathrm{O}}$ caused by spurious global ocean mass changes in OCCIPUT under the Boussinesq approximation. Legitimate contributions to the globally averaged $p_{\mathrm{b}}$, which would be removed by the correction, pertain to the freshwater flux into the ocean from atmospheric and continental sources. However, this term is accounted for

through the framework of GAL, and thus in $\hat{\chi}^{\mathrm{L}}$.



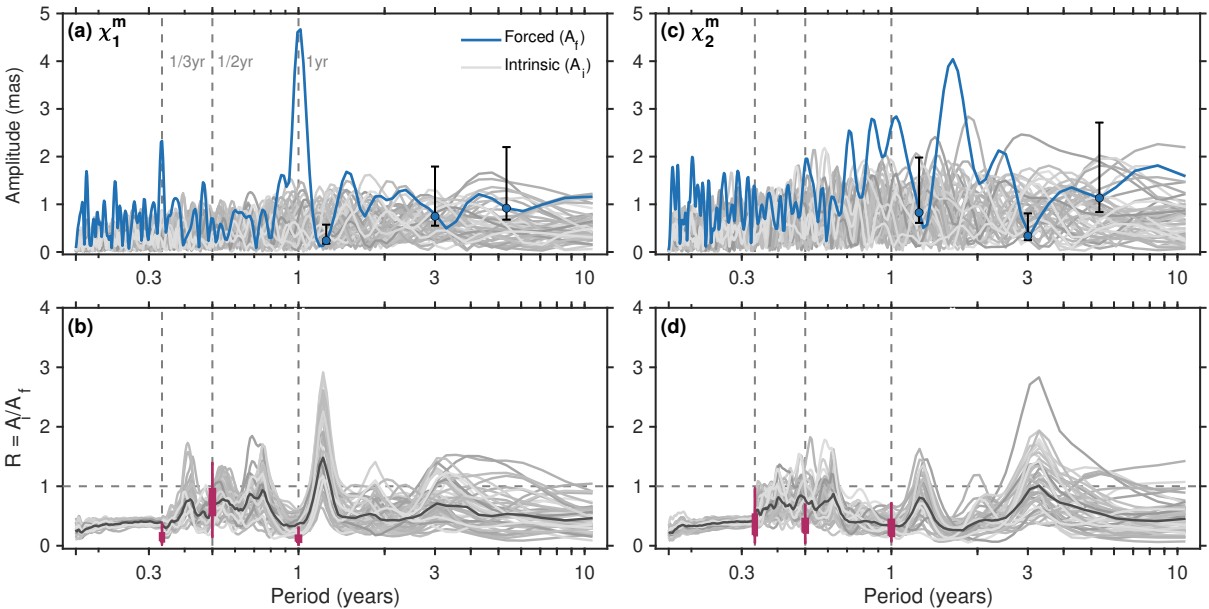

**Figure 2.** Spectral representation of the OCCIPUT-based estimates of intrinsic and forced oceanic excitation of polar motion, 1995–2015. Shown are amplitudes $A_i$ and $A_f$ (in milliarcseconds) of the intrinsic and forced oceanic mass term, $\hat{\chi}^m$, in (a) $\chi_1$ and (c) $\chi_2$, deduced from a 512-point Fast Fourier Transform using Welch's method (Welch, 1967). Black vertical bars indicate 68% confidence limits for selected frequencies. Panels (b, d) illustrate spectrally smoothed versions of the chaotic amplitude fraction $R = A_i/A_f$ (dimensionless), estimated from the excitation time series with seasonal terms removed. Black solid lines mark the respective median across the 50 realizations of $R$. Statistics for the annual, semiannual, and terannual terms—computed separately based on seasonal cycle fits to $\hat{\chi}^{O,i}$ and $\hat{\chi}^{O,f}$—are summarized by the purple whiskers; box edges indicate the 25th and 75th percentiles and vertical bars extend to the minimum/maximum values of seasonal $R$.

## 3    Results and discussion

### 3.1    Overview in spectral domain

A natural first step is to assess the magnitude and importance of the intrinsic oceanic excitation in relation to the externally forced component across a range of frequencies. In Figs. 2 and 3, we show amplitude spectra (symbol $A$) for $\hat{\chi}^{O,i}$ (50 real-

izations) and $\hat{\chi}^{O,f}$ (1 realization), separated into mass and motion terms and the two coordinate directions. For comparison purposes, we form chaotic amplitude fractions, $R = A_i/A_f$, representing the ratio of intrinsic over forced amplitude per frequency (not to be mixed up with time-invariant fractions of variance considered in previous OCCIPUT analyses, e.g., Carret et al., 2021; Hogg et al., 2022). Values of $R$ are computed separately for (a) annual, semiannual, and terannual harmonics extracted from $\hat{\chi}^{O,i}$ and $\hat{\chi}^{O,f}$, and (b) the deseasonalized excitation functions. We smooth the estimated fractions under (b) using

a moving window, with window sizes decreasing (essentially linearly) from 21 frequency bins at the high end of the spectrum to 1 bin at the longest period.





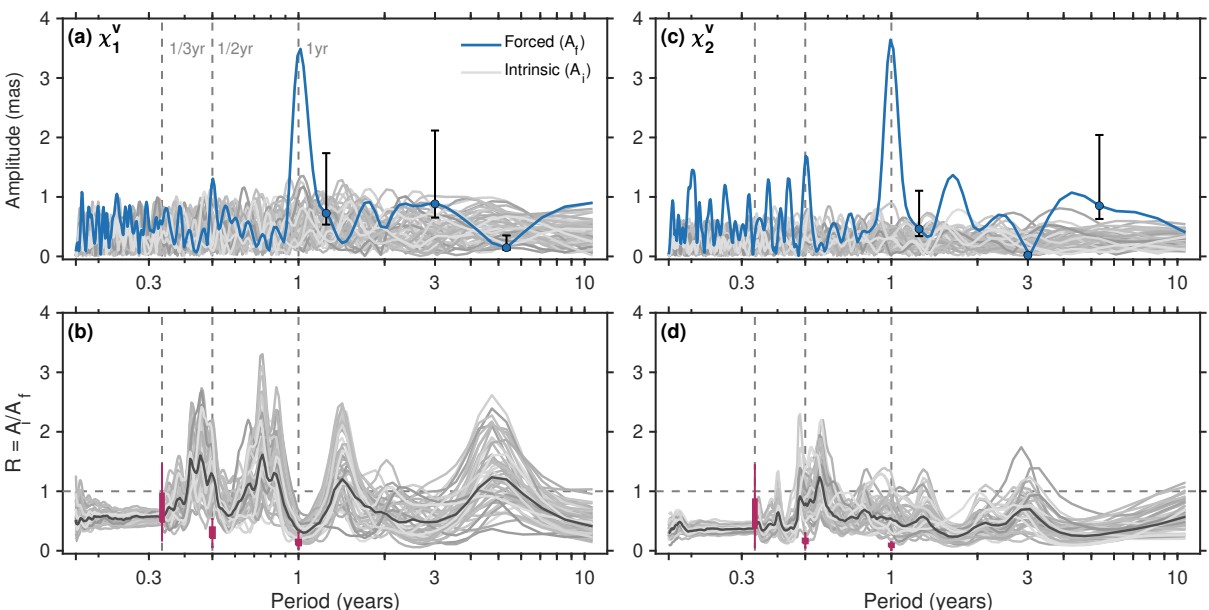

**Figure 3.** As in Fig. 2 but for the two components of the motion term $\hat{\chi}^{\mathrm{v}}$.

As apparent from Figs. 2 and 3 (panels a/c), the intrinsic oceanic excitation is embedded in a relatively flat spectrum, bounded by amplitudes of $\sim$1.5–2.0 mas in the mass term and $\sim$0.7–1.0 mas in the motion term. This spectral characteristic renders the chaotic excitation comparable in magnitude to the forced variability at most, but not all frequencies. Notable exceptions

occur in $\chi_2^{\mathrm{v}}$ and wherever $\hat{\chi}^{\mathrm{O,f}}$ features peaks in seasonality. Values of $R$ are close to 0.15 for the dominant ($\sim$3.5 mas) annual harmonic in both $\chi_1^{\mathrm{v}}$ and $\chi_2^{\mathrm{v}}$, along with similarly small fractions at periods of 1 and 1/3 yr in $\chi_1^{\mathrm{m}}$. We can tentatively interpret the results for $\chi_1^{\mathrm{m}}$ in terms of intrinsic-vs-forced ratios of $p_{\mathrm{b}}$ signals from previous OCCIPUT analyses (Zhao et al., 2021). In particular, Fig. 3 of Zhao et al. (2021) places large intrinsic contributions to the seasonal $p_{\mathrm{b}}$ cycle over western boundary currents and the eddying Southern Ocean, with little sign of long-wavelength effects that would drive strong variability in $\hat{\chi}^{\mathrm{O,i}}$.

Switching from seasonal harmonics to the broadband spectral behavior in Figs. 2 and 3, we observe a gradual increase in the chaotic amplitude fraction from $\sim$0.3 to $\sim$0.6 as periods lengthen from 60 to 120 days. This is followed at intraannual frequencies by somewhat of a plateau of the median $R$ near 0.9 in the mass term, along with a broadening of the ensemble spread to $R \approx 0.3$–2.0 in all OAM components. The prominence of $\hat{\chi}^{\mathrm{O,i}}$ at these periods (roughly 120 to 300 days) coincides with the dominant timescale for emergence of mesoscale eddies (Chelton et al., 2011) and the ensuing nonlinear inverse

cascades (e.g., Arbic et al., 2014; O'Rourke et al., 2018; Sérazin et al., 2018). Concurrently, the forced oceanic excitation is relatively weak at intraannual periods (especially in $\chi_1^{\mathrm{O}}$, Figs. 2a and 3a), making this specific band another attractive (future) target for exploring the polar motion imprint of oceanic chaos. We note, though, that the intraannual wobble excitation is largely dominated by atmospheric mass redistribution effects (Gross et al., 2003).





For the interannual signals central to this work, the spectral representations in Figs. 2 and 3 indicate appreciable power in

$\hat{\chi}^{O,i}$ relative to $\hat{\chi}^{O,f}$, even at sub-decadal periods and particularly in $\chi_1^O$. Some low-frequency, intrinsic oceanic process must

thus be at work at scales large enough to perturb the globally integrated OAM quantities (see Sect. 3.2). More specific to Figs. 2

and 3, we observe a number troughs in the forced amplitude $A_f$ that in turn afford peaks in the chaotic amplitude fractions.

Such maxima in $R$ are seen, for example, for periods $T \approx 1.2$–$1.3$ years in all four $\hat{\chi}^O$ components or near 3 years in the two

mass terms (spectral uncertainty is generally large, though). By contrast, the forced component exceeds the intrinsic amplitude

by a factor of about two at $T \approx 1.6$ years in both mass and motion terms of $\chi_2^O$. One can therefore expect any time series of

interannual $\chi_2^O$ signals to feature stronger deterministic variability than $\chi_1^O$.

For a condensed view of Figs. 2 and 3, we have additionally formed ratios between intrinsic and forced standard deviations,

i.e., $\sigma_i/\sigma_f$, calculated from $\hat{\chi}^{O,i}$ and $\hat{\chi}^{O,f}$ time series. Summing up mass and motion terms, and filtering the participating time

series to periods $\geq 14$ months, we obtain $\sigma_i/\sigma_f$ values of 0.50–0.83 (median 0.69) for $\chi_1^O$ and 0.28–0.50 (median 0.36) for

$\chi_2^O$. Similar calculations for the complex-valued quantity $\hat{\chi}^O$ yield $\sigma_i/\sigma_f$ ratios in the range of 0.38–0.60 and a median at 0.47.

When taking $\sigma_i$ relative to the variability of the total oceanic excitation, the median is 0.46 in the case of $\hat{\chi}^O$ (range 0.43–0.50).

These numbers emphasize the importance of chaotic contributions to equatorial OAM changes on time scales much longer

than typically associated with the life cycles of mesoscale eddies.

### 3.2    Modal decomposition

Analysis of the space-time variability in modeled intrinsic $p_b$ fields—denoted as $p_b^i$—can reveal more about the provenance

of the chaotic OAM signals apparent in Figs. 2 and 3. Following Zhao et al. (2023), we have decomposed each of the 50

realizations of $p_b^i$ into real-valued EOF patterns and the associated principal components. We find a common leading EOF

(mode 1 hereafter), accounting for 16% to 22% of the global $p_b^i$ variance. This particular mode, displayed exemplarily for

one ensemble member in Fig. 4a, exhibits a distinct global structure, characterized by covariability across the Atlantic, the

Arctic, and partly the Indian Ocean, while much of the Southern Ocean oscillates with opposite sign. The suggested origin for

this mode lies in Drake Passage (Zhao et al., 2023). Here, strong eddies and random current fluctuations, possibly involving

interaction with topography (Provost et al., 2011), are thought to induce sharp bottom pressure gradients, evident in Fig. 4a

between Cape Horn and the South Shetland Islands. The basin-wide $p_b^i$ anomalies seen elsewhere in the ocean would then be

the result of fast barotropic adjustment to these local gradients along $H/f$ contours (Zhao et al., 2023, where $H$ is the local

water depth, and $f$ is the Coriolis parameter).

The described processes, while of different nature than the nonlinear inverse energy cascades noted in Sect. 1, are unique

in that they produce a global pattern of mass redistribution with opposite polarity between ocean basins. Such large-scale see-

saws of mass have been previously shown to be an effective source of polar motion excitation; cf., e.g., Afroosa et al. (2021).

We have thus synthesized monthly $p_b^i$ grids from mode 1 alone and deduced the corresponding oceanic excitation (that is,

the oceanic mass term, Appendix A). The so-derived estimates of $\hat{\chi}^{O,i}$ are compared in Table 1 against the full $\hat{\chi}^{O,i}$ functions

from OCCIPUT in terms of the percentage of variance explained (PVE). This simple evaluation demonstrates that mode 1

is indeed the leading driver for the mass component of intrinsic oceanic excitation, across time scales and for both $\chi_1$ and



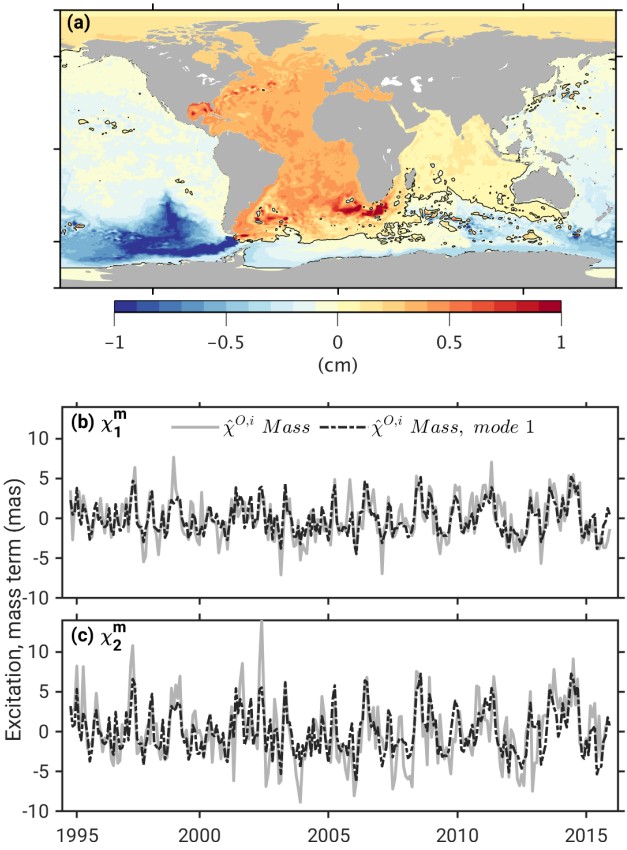

**Figure 4.** Attribution of the mass term of $\hat{\chi}^{O,i}$ to global-scale intrinsic bottom pressure variability, $p_b^i$. Shown is (a) the leading EOF mode of $p_b^i$ variability (in centimeter) for ensemble member 47, and (b, c) the corresponding equatorial oceanic excitation signal (dashed black curve), obtained from synthesizing mode 1 in time and space and computing the associated OAM time series. The full intrinsic oceanic excitation, derived from $p_b^i$ without modal partitioning, is plotted in gray. We selected member 47 for this comparison as its statistics in Table 1 are close to the median percentage of variance explained by mode 1 in $\hat{\chi}^{O,i}$. Note that other ensemble members suggest a mode 1 structure very similar to the one shown in panel (a); see also Zhao et al. (2023).

$\chi_2$. Specifically, at the interannual periods of interest in this work, the OAM changes associated with $p_b^i$ mode 1 explain 39% to 97% of the variance in the $\hat{\chi}^{O,i}$ mass term, with a median PVE of 63% across the 50 ensemble members. These numbers encapsulate the idea that localized mesoscale processes somewhere in the world ocean govern variability in a truly global and geophysically relevant quantity.

### 3.3 Interannual variability

We now examine the role of oceanic chaos in the low-frequency equatorial excitation budget over 1995–2015. To that end, Fig. 5 shows the residual geodetic excitation, obtained by subtracting from $\hat{\chi}^*$ modeled contributions from the atmosphere,





**Table 1.** Contribution of mode 1 $p_b^i$ variability to the intrinsic component of the oceanic mass term, 1995–2015[a]

|  | $\chi_1$ | $\chi_2$ | $\hat{\chi} = \chi_1 + i\chi_2$ |
|---|---|---|---|
| PVE in $\chi_j^{O,i}$ by EOF mode 1 | | | |
| min | 38.4 | 35.0 | 40.4 |
| median | 56.6 | 49.6 | 52.3 |
| max | 97.4 | 97.9 | 97.7 |
| PVE in low-frequency $\chi_j^{O,i}$ by EOF mode 1[b] | | | |
| min | 35.6 | 28.3 | 39.0 |
| median | 67.8 | 63.9 | 63.3 |
| max | 96.7 | 97.1 | 96.8 |

[a] Values are PVE by a monthly-varying $\hat{\chi}^{O,i}$ mass term computed from mode 1 vs. the full intrinsic mass term. Subscript $j$ is used to discriminate between $\chi_1$, $\chi_2$, and $\hat{\chi}$.

[b] Participating time series filtered to periods $\geq 14$ months.

terrestrial hydrology, the two ice sheets, and GAL; cf. the respective time series in Fig. 1. Together, these non-oceanic sources account for 29.7% of the variance in the full equatorial geodetic excitation (Table 2), governed in large parts by the PVE in $\chi_1$ (51.6%). The residual is to be compared with 51 variants of oceanic excitation from OCCIPUT, comprising both the forced component ($\hat{\chi}^{O,f}$) and 50 realizations of $\hat{\chi}^O$ (i.e., $\hat{\chi}^{O,f} + \hat{\chi}^{O,i}$). Intrinsic processes clearly add extra variability to estimates of $\hat{\chi}^O$ relative to $\hat{\chi}^{O,f}$, as conveyed by an ensemble spread of nearly $\pm 5$ mas throughout the analysis period. This is an important

result that implies some leeway in how to evaluate and interpret the observed excitation of Earth's wobbles on interannual time scales.

Adopting the forced component from OCCIPUT as a middle ground for an estimate of $\hat{\chi}^O$ reduces the RMS (root-mean-square) of the residual equatorial geodetic excitation from 9.6 mas to 6.7 mas (PVE = 50.8%). Similarly, we can select from any of the 50 ensemble members as valid representations of the time-evolving ocean state and thus $\hat{\chi}^O$. This approach to

evaluation leads to a considerable spread in the statistics in Table 2, involving a few ensemble members that exhibit slightly better skill than the forced component (minimum $\hat{\chi}$ RMS of 6.5 mas, PVE = 53.7%), but also a number of cases that reduce little variance (only 1.5 mas in RMS) of the residual geodetic excitation. A total of 15 members even yield negative PVE values in $\chi_1$. However, these statistics are not to be taken to define absolute bounds on the efficacy of OAM in exciting low-frequency polar motion, as the intrinsic variability in any of the ensemble members is merely a random realization of the potential intrinsic

variability in the real world. In fact, oceanic chaos may act to produce yet other time series of $\hat{\chi}^O$ that are also consistently close to the residual geodetic excitation, while still lying within the ensemble spread in Fig. 5.

We have constructed such a hypothetical—but still legitimate—equatorial OAM function by identifying, month per month and in the complex plane, the particular OCCIPUT $\hat{\chi}^O$ sample closest to the reduced observation (i.e., $\hat{\chi}^*$ minus non-oceanic

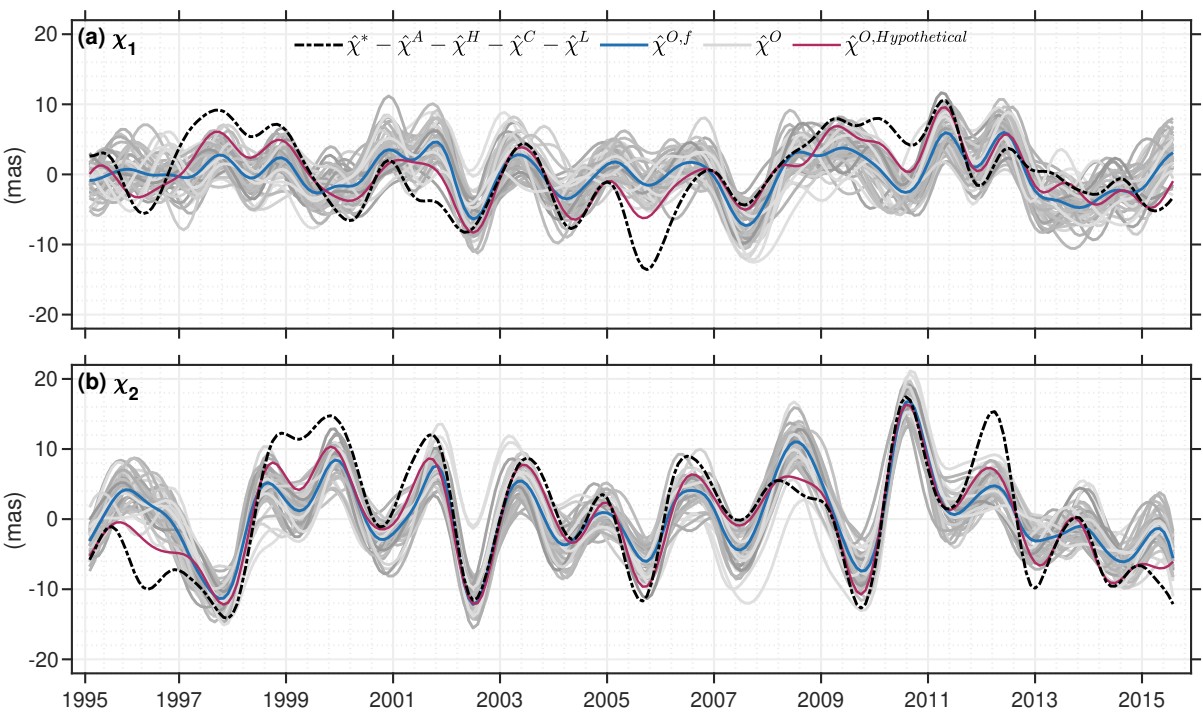

**Figure 5.** The low-frequency polar motion excitation budget from 1995 to 2015 in the presence of oceanic chaos. Shown are (as dashed black lines) the (a) $\chi_1$ and (b) $\chi_2$ components of geodetic excitation ($\hat{\chi}^*$)—corrected for contributions from the atmosphere ($\hat{\chi}^A$), terrestrial hydrology ($\hat{\chi}^H$), ice sheets ($\hat{\chi}^C$), and GAL ($\hat{\chi}^L$)—compared with 50 estimates of oceanic excitation from OCCIPUT ($\hat{\chi}^O$, gray curves), the OCCIPUT ensemble mean ($\hat{\chi}^{O,f}$, blue curve), and a hypothetical ensemble member (purple curve) producing the best match with the corrected geodetic excitation. All time series have been detrended, filtered to periods $T \geq 14$ months, and cut back by 4 months at the two end points.

sources). As one switches from one low-pass filtered member to the other, small jumps are incurred, requiring an additional
round of smoothing to periods $\geq 14$ months. As per design, the resulting time series follow the residual geodetic excitation more tightly than $\hat{\chi}^{O,f}$ or any of the ensemble members (Fig. 5). The improved correspondence is also reflected in the PVE by the hypothetical $\hat{\chi}^O$ (84.0% of the reduced geodetic excitation), leaving only 3.8 mas of interannual RMS variability unexplained. Likewise, using the hypothetical member instead of the ensemble mean in evaluating the sum of all geophysical fluid excitations against $\hat{\chi}^*$ increases the PVE from 65.4% to 89.0%. For comparison, selecting from the original OCCIPUT members to form
the total surface layer excitation yields PVE values from 50.2% to 67.5%. Overall, these statistics illustrate the sensitivity of such budget analysis to the representation of chaotic OAM signals.

As evident from Fig. 5, the 3.8 mas RMS signal unaccounted for in the observations is mostly made up by discrepancies between geodetic and geophysical excitations before early 2002 (i.e., the launch of GRACE), large differences throughout the year 2005 in $\chi_1$ (to a lesser extent also 2009/2010), and a ∼7 mas cusp around the turn of 2011/2012 in $\chi_2$. It is possible that part





**Table 2.** Observed vs. modeled excitation of interannual polar motion, 1995–2015[a]

|  | $\chi_1$ | $\chi_2$ | $\hat{\chi} = \chi_1 + \mathrm{i}\chi_2$ |
|---|---|---|---|
| RMS of $\chi_j^*$ | 7.5 | 8.6 | 11.4 |
| PVE in $\chi_j^*$ by non-oceanic sources |  |  |  |
| $\chi_j^A + \chi_j^H + \chi_j^C + \chi_j^L$ | 51.6 (5.2) | 13.3 (8.0) | 29.7 (9.6) |
| PVE in $\chi_j^*$ by OCCIPUT, $\chi_j^O$ |  |  |  |
| Forced | 18.5 (6.8) | 35.4 (6.9) | 28.2 (9.7) |
| Ensemble members | [−17.5–27.4] ([6.4–8.1]) | [10.2–44.7] ([6.4–8.2]) | [5.3–34.2] ([9.3–11.1]) |
| PVE by OCCIPUT $\chi_j^O$ in residual series $\chi_j^* - \chi_j^A - \chi_j^H - \chi_j^C - \chi_j^L$ |  |  |  |
| Forced | 23.9 (4.5) | 62.1 (5.0) | 50.8 (6.7) |
| Ensemble members | [−24.4–35.1] ([4.2–5.8]) | [44.8–64.6] ([4.8–6.0]) | [29.1–53.7] ([6.5–8.1]) |
| Hypothetical member[b] | 77.4 (2.5) | 87.4 (2.9) | 84.0 (3.8) |
| PVE in $\chi_j^*$ by sum of excitation processes, $\chi_j^A + \chi_j^O + \chi_j^H + \chi_j^C + \chi_j^L$ |  |  |  |
| Forced | 63.1 | 67.1 | 65.4 |
| Ensemble members | [39.8–68.6] | [52.1–69.3] | [50.2–67.5] |
| Hypothetical member[b] | 89.0 | 89.1 | 89.0 |

[a] Values are PVE and the corresponding RMS of residuals (milliarcseconds) in parentheses, except for the first line. Entries for ensemble members apply to the sum of the forced and intrinsic signals. All time series have been detrended and filtered to periods $T \geq 14$ months. Subscript $j$ in the intermediate headers is used to discriminate between $\chi_1$, $\chi_2$, and $\hat{\chi}$.

[b] Hypothetical member drawn from the $\hat{\chi}^O$ ensemble based on a constraint of minimum distance to the residual observed excitation, see the main text.

of the discrepancies arise from errors in OCCIPUT, related to, e.g., imperfectly modeled energetic circulations in the Southern Ocean (Ponte and Piecuch, 2014; Harker et al., 2021). A more likely source, though, are residual errors in the adopted gravity fields that project onto the interannual excitation budget, particularly through TWS changes and the associated fluctuations in $\hat{\chi}^H$. Comparing our hydrological time series in Fig. 1 with GRACE-only estimates of $\hat{\chi}^H$ in Adhikari and Ivins (2016) (their Fig. 3) does indeed point to differences over the afore-mentioned periods (2005 in $\chi_1$, 2011/2012 in $\chi_2$). It therefore appears

that the specifics of the gravity field reconstruction—including the limited spectral sensitivity of the involved satellites and the temporal parameterization in terms of GRACE/-FO EOFs (Löcher and Kusche, 2021)—occasionally compromise the accuracy of the derived hydrological angular momentum changes. More detailed exploration of these factors is left for future studies.

Although the geophysical models and data used in this study are necessarily imperfect, it is worth inspecting the excitation budget from yet another angle. In Fig. 6 we show amplitude spectra for the two components of $\hat{\chi}^*$ minus all climate system-

related excitations, including for present purposes also the forced oceanic component ($\hat{\chi}^{O,f}$). Spectral power in these residuals—while not well resolved within our 21-year analysis window—may be due the said errors in the adopted angular momentum





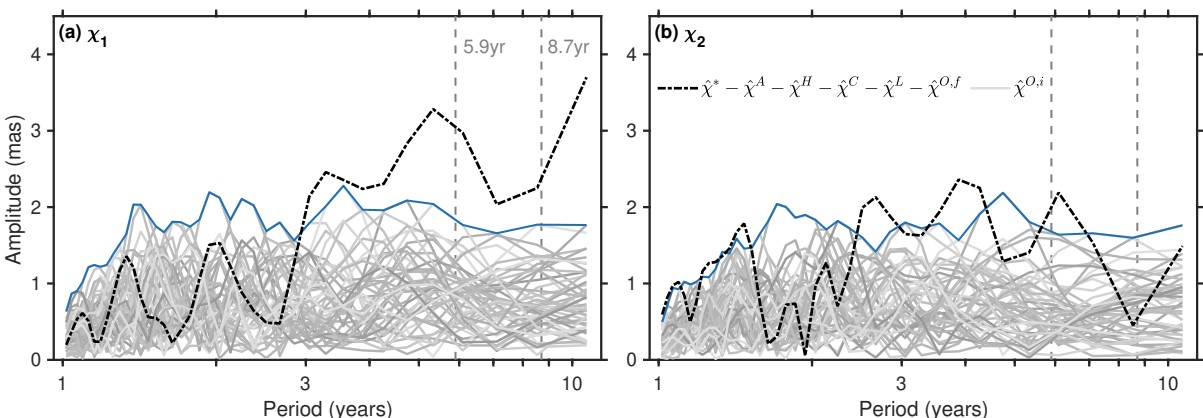

**Figure 6.** Amplitude spectra (in milliarcseconds) of $\hat{\chi} = \chi_1 + \mathrm{i}\chi_2$ over 1995–2015 for the intrinsic oceanic excitation (50 realizations of $\hat{\chi}^{O,i}$, gray lines), in comparison to the residual geodetic excitation (black-dotted line), obtained by subtracting from $\hat{\chi}^*$ estimated contributions from atmosphere, terrestrial hydrology, ice sheets, GAL, and the forced oceanic excitation ($\hat{\chi}^{O,f}$); cf. the expression in the legend. The upper envelope of the intrinsic terms is represented by the blue line.

series. Alternatively, residual power may point to secondary excitation processes left out of the considerations and coming from, e.g., the Earth's core (Chen et al., 2019; Ding et al., 2019; Kuang et al., 2019). Inferences along the latter lines are however complicated by the difficulty in accounting for the intrinsic component of oceanic excitation, $\hat{\chi}^{O,i}$, as expressed by the gray background spectra in Fig. 6. In detail, the OCCIPUT ensemble suggests a likelihood for $\hat{\chi}^{O,i}$ to attain amplitudes between 0 and 1.5 mas across all interannual and sub-decadal frequencies; cf. also Figs. 2 and 3. Selecting always the most energetic ensemble member at each frequency (blue lines in Fig. 6) raises the upper bound for intrinsic effects even further, to about 2 mas.

A 1.0–1.5 mas uncertainty in modeled wobble excitation due to oceanic chaos is generally comparable in size to the residual geodetic excitation across most frequencies (black dashed curves in Fig. 6). This prompts the question as to what extent it is possible to conclusively link such residuals to the dynamics of Earth's deep interior. For instance, Chen et al. (2019) suggested that a ∼2 mas oscillation with a period of $T = 5.9$ years—left after correcting the observed excitation over 1962–2018 for modeled atmospheric, oceanic, and hydrological effects—may be caused by the exchange of equatorial angular momentum between the outer core and the mantle via the electromagnetic torque (Kuang et al., 2019). However, analyses of satellite-derived degree-2 Stokes coefficients (Rosat et al., 2021) fails to support this idea, while our Fig. 6 illustrates that a considerable portion of any interannual $\hat{\chi}^*$ residual may simply be due to mis-modeled oceanic effects. Furthermore, the possibility of intrinsic OAM signals exciting polar motion at the level of 1.0–1.5 mas even at sub-decadal periods ($T = 5$–10 years) adds uncertainty to attempts of detecting the inner core wobble (ICW, Rochester and Crossley, 2009) from rotation data. Recent studies (e.g., Ding et al., 2019) have made the case for a 8.7-year prograde harmonic in residual polar motion to be the signature of the ICW, at an amplitude of just $2.67 \pm 0.04$ mas. However, the OAM product used by Ding et al. (2019) to remove oceanic contributions from the observations did not account for intrinsic variability, implying that the error bars on the inferred





ICW amplitude are likely too optimistic. Substantiating the evidence for core effects in polar motion data on sub-decadal time scales will most certainly require further improvements in modeling surface layer processes, comprising in particular oceanic and hydrological angular momentum changes (Rosat et al., 2021; Rekier et al., 2022).

## 4 Summary and conclusions

The 50-member OCCIPUT ensemble investigated in this study advocates for a prominent role of oceanic intrinsic variability in equatorial OAM changes from intraseasonal to interannual time scales. This marks the arrival of a new player on the scene of Earth rotation variations, with implications for tests of closure in the polar motion excitation budget and yet unmodeled geophysical processes (Table 2 and Fig. 6). In terms of attribution, we have been able to relate most of the mass-driven signals in $\hat{\chi}^{\mathrm{O,i}}$ (median PVE $= 63\%$ at periods $\geq 14$ months) to one particular mode of intrinsic $p_{\mathrm{b}}$ variations that likely manifests the free barotropic adjustment of the main ocean basins to nonlinearly-induced $p_{\mathrm{b}}$ anomalies near Drake Passage (cf. Fig. 4 and Zhao et al., 2023). As in many other parts of the Southern Ocean, low-frequency changes in eddy activity around Drake Passage are strongly random in character (Hogg et al., 2022), thus providing a plausible explanation as to why variability in $\hat{\chi}^{\mathrm{O,i}}$ is a considerable fraction (43–50%, see Sect. 3.1) of the total oceanic excitation of polar motion even on interannual time scales. A caveat to be acknowledged, though, is that all of these inferences are based on simulations. The credibility of OCCIPUT-based OAM series is however clear from Table 2, and comparisons with monthly GRACE/-FO fields have shown that the leading $p_{\mathrm{b}}^{\mathrm{i}}$ mode inherent to the ensemble could be present in satellite observations (Zhao et al., 2023). In addition, the Southern Ocean eddy field in the 1/4° OCCIPUT runs is slightly less intense than in altimetry data (Carret et al., 2021; Hogg et al., 2022), suggesting that our estimates for the chaotic contribution to wobble excitation are rather of conservative nature.

Our results underscore the challenge of accurately representing oceanic effects for polar motion studies. In general, high horizontal model resolution ($\leq 1/4°$) is preferred in accounting for wind-driven OAM changes (Harker et al., 2021). However, a single, high-resolution baroclinic model run would also produce one specific realization of the intrinsic component that is unlikely to match the phase of the observed one. Conversely, pursuing a probabilistic approach with a large eddy-permitting ensemble such as OCCIPUT may be computationally impracticable, especially on an operational basis and at the (sub-)daily resolution now common for rotation studies (Dobslaw and Dill, 2018). Better constraining the chaotic component of $\hat{\chi}^{\mathrm{O,i}}$ will most certainly require an eddying ocean model assimilating, amongst other observations, altimetric sea level anomalies and GRACE/-FO estimates of $p_{\mathrm{b}}$ (i.e., extensions of the ocean reanalyses analyzed in Börger et al., 2023). In the absence of such frameworks, the ensemble spreads and statistics presented in the present work provide indications of the errors incurred by methods that neglect intrinsic OAM variations. We expect these error quantifications to be useful for prediction purposes (e.g., Dobslaw and Dill, 2018; Kiani Shahvandi et al., 2022) and the geophysical interpretation of the residual geodetic excitation of polar motion from intraseasonal to sub-decadal time scales.



*Data availability.* All geophysical angular momentum estimates analyzed in this study have been placed at https://doi.org/10.5281/zenodo .12664036 (Börger et al., 2024).

## Appendix A: Effective angular momentum functions

To estimate the effect of fluid angular momentum changes on low-frequency polar motion, we use the frequently cited formalism of Gross (2007)

$$\hat{\chi} \;=\; \hat{\chi}^{\mathrm{m}} + \hat{\chi}^{\mathrm{v}} = \frac{1.100\,\Omega\,\Delta\hat{I} + 1.608\,\hat{h}}{\Omega(C - A')} \;. \tag{A1}$$

Here, $\Omega$ ($7.292115\cdot10^{-5}$ rad s$^{-1}$) is Earth's nominal sidereal rotation rate, $C$ ($8.0365\cdot10^{37}$ kg m$^2$) and $A'$ ($8.0102\cdot10^{37}$ kg m$^2$) are the polar and average equatorial principal moments of inertia of the Earth (including the core), $\hat{h} = h_1 + \mathrm{i}h_2$ represents the

relative angular momentum of the considered fluid, and $\Delta\hat{I} = \Delta I_{13} + \mathrm{i}\Delta I_{23}$ is the time-dependent equatorial perturbation to third-column elements in the tensor of inertia (Moritz and Mueller, 1987). The numerical coefficients account for the responses of the anelastic mantle, fluid core, and equilibrium ocean to rotational perturbations caused by the initial excitation, and also the deformation of the solid Earth due to surface mass loading. Implicit to Eq. (A1) is the assumption of no mechanical coupling between mantle and core during polar motion fluctuations. The validity of this assumption on time scales longer than the

Chandler period is somewhat contested (Dickman, 2003), but geophysical arguments (Wahr, 2005) suggest that even complete core-mantle coupling would modulate the transfer coefficients by no more than a few percent.

Writing $\Delta\hat{I}$ and $\hat{h}$ in spherical polar coordinates, we have, for the ocean (cf. Gross et al., 2003)

$$\Delta\hat{I} \;=\; -\frac{a^4}{g} \int\limits_{-\frac{\pi}{2}}^{\frac{\pi}{2}} \int\limits_{0}^{2\pi} p_{\mathrm{b}} \sin\phi \cos^2\phi \, \exp(\mathrm{i}\lambda) \, \mathrm{d}\lambda\,\mathrm{d}\phi \tag{A2}$$

$$\hat{h} \;=\; -a^3\rho_0 \int\limits_{-\frac{\pi}{2}}^{\frac{\pi}{2}} \int\limits_{0}^{2\pi} (\overline{u}H \sin\phi + \mathrm{i}\overline{v}H) \cos\phi \, \exp(\mathrm{i}\lambda)\,\mathrm{d}\lambda\,\mathrm{d}\phi \;, \tag{A3}$$

where $a$ is Earth's mean radius, $\lambda$ and $\phi$ are longitude and latitude of an arbitrary point on the Earth, $g$ is the constant gravitational acceleration, $\rho_0$ is a mean seawater density (1025 kg m$^{-3}$), and $H$ denotes the local water depth. The quantities $(p_{\mathrm{b}}, \overline{u}, \overline{v})$ are the dynamical variables specified in Sect. 2.2, representing either forced or intrinsic variations, or their sum. Expressions analogous to Eq. (A2) exist for the mass term of other Earth system components, with pressures readily computed from equivalent water heights via the hydrostatic equation. Apart from the ocean, only the atmosphere has a non-zero motion term, usually

evaluated as volume integral over horizontal currents and densities in three-dimensional space (Barnes et al., 1983).

## Appendix B: Terrestrial hydrology and cryosphere

The monthly gravity fields used for the determination of $\hat{\chi}^{\mathrm{H}}$ and $\hat{\chi}^{\mathrm{C}}$ over the 1995–2015 period are computed from tracking observations to up to 16 satellites from both satellite laser ranging (SLR) and the Doppler Orbitography and Radiopositioning



Integrated by Satellites (DORIS) system. The problem is treated in terms of variational equations and cast as an iterative

fit of forward-modeled satellite orbits to the observations. In representing the gravity field, low-degree spherical harmonics
are combined with EOFs derived by principal component analysis of GRACE/-FO gravity fields. This hybrid representation,
developed in Löcher and Kusche (2021), allows for the same effective wavelengths as the GRACE/-FO solutions while reducing
the number of parameters to an amount appropriate for techniques less sensitive to gravity field details. As the approach leaves
some scope for the choice of the base functions, the monthly fields in this study are computed as weighted means of 15 solutions

employing various numbers of EOFs (up to 14) and spherical harmonics (from degree 2 to 5).

The coordinates of the SLR and DORIS stations entering the solution are part of the International Reference Frame 2014
(ITRF2014, Altamimi et al., 2016), corrected for tidal and non-tidal loading as appropriate. The force modeling for all satellites
accounts for pole tides as per conventions (Petit and Luzum, 2010) and ocean pole tides following Desai (2002). Short-term
mass variations in atmosphere and ocean are removed via the GRACE dealiasing product AOD1B RL06 (the GAC coefficients,

Dobslaw et al., 2017) except for degree-1 terms. Furthermore, the force model includes the gravitational effect of glacial
isostatic adjustment (GIA) based on the model of A et al. (2013); choice of another GIA estimate has negligible effects on
our polar motion results. For consistency, we also remove the A et al. (2013) model from the GRACE/-FO solutions prior to
expanding them into EOFs. Neither GIA nor the AOD1B product is restored to the final gravity fields.

In a subsequent step, we also deduce degree-1 coefficients (i.e., geocenter displacements), by keeping the gravity field

invariant and estimating station coordinates for all SLR sites using a no-net-rotation condition. The common mode of translation
across the so-derived stations reveals the time-variable offset of the geocenter relative to the ITRF2014. Over the GRACE/-
FO time span, our degree-1 time series is very similar to, but somewhat noisier than the widely used estimate of Swenson
et al. (2008). Prior to assigning this geocenter model to the monthly solutions, we subtract degree-1 contributions from the
atmosphere (based on ERA-Interim with IB) and the ocean (based on forced OCCIPUT $p_b$). This correction avoids double-

counting of long-wavelength mass variations in our analysis, associated in particular with atmospheric pressure changes over
land.

The recovered gravity fields inherit the directional error structures (stripes) from the GRACE/-FO EOFs and thus require
filtering in post-processing. We use a standard decorrelation and anisotropic smoothing kernel (DDK2, Kusche et al., 2009)
to that end. Each monthly set of Stokes coefficients (including degree-1 terms) is then mapped to gridded ($1° \times 1°$) surface

mass anomalies and iteratively corrected for signal leakage across continental boundaries into the ocean. Our approach closely
follows Chen et al. (2015). In short, the algorithm (a) takes a first guess of the true, gridded mass distribution, (b) derives
a "predicted" mass field from (a) by expanding and re-synthesizing it using spherical harmonics up to degree 60, and (c)
successively corrects the prediction—and thus the current guess for the true mass distribution—to yield closer agreement with
the particular month's observed mass anomalies. The latter also defines the first guess but with values over the ocean replaced

by their spatial mean. This delineation leads to a sharp transition between the ocean and land (or ice sheets, respectively) that
is kept for all guesses in the iteration. We separately iterate land, Greenland, and Antarctica, always permitting adjustments to
local mass values within 600 km from the coast and correcting the spatial mean over the ocean accordingly to conserve total



mass. Acceptable convergence is typically achieved after five iterations per modeled region. The resulting mass change fields, used as inputs for Eq. (A2), are illustrated in terms of their standard deviation in Figs. B1a–B1c.

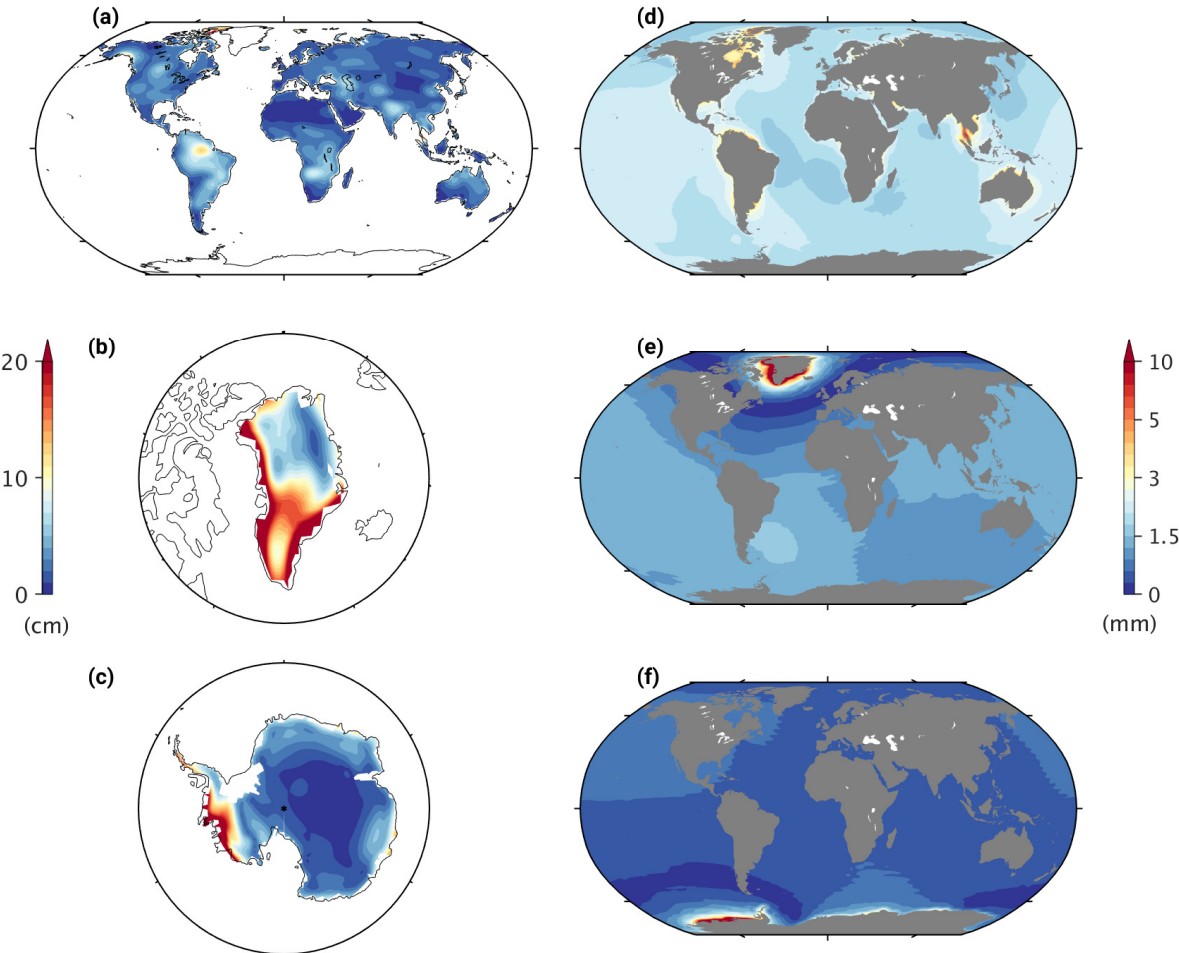

**Figure B1.** Standard deviation of (a–c) interannual surface mass changes over continents, as deduced from long-term gravity field solutions, and (d–f) associated GAL-induced mass redistribution in the ocean, from 1995 to 2015. Mass and GAL signals are depicted separately for (a,d) land hydrology, (b,e) Greenland, and (c,f) Antarctica. Units are centimeter of equivalent water height. Trends, seasonal oscillations, and sub-annual signals have been removed from each grid point's time series and therefore do not contribute to the picture.

**Appendix C: Gravitational attraction and loading**

To quantify the effect of surface mass loads on polar motion excitation, we combine mass load estimates from different sources, generally separated by Earth system compartment. For a precise net excitation estimate, allowance should be made for trans-





ports of water between continents and the ocean, leading to the residual excitation $\hat{\chi}^{\mathrm{L}}$. We require our estimate of $\hat{\chi}^{\mathrm{L}}$ to be consistent with the physics of GAL, such that the total load mass is conserved and the modeled sea level is an equipotential of

Earth's gravity field (Nakiboglu and Lambeck, 1980; Quinn et al., 2015; Adhikari and Ivins, 2016). Under the assumption that these mass fluxes and gravitational effects cause no relevant dynamic variations in the ocean (e.g., Kuhlmann et al., 2011), the resultant non-uniform water mass displacements may be estimated by solving the sea level equation (Woodward, R. S., 1888; Farrell and Clark, 1976). Here, sea level is taken relative to the deformed crust, equivalent to a change in $p_{\mathrm{b}}$ in the absence of atmospheric pressure variations. We solve the sea level equation in the spectral domain based on a global fingerprint inver-

sion framework (Rietbroek et al., 2016; Uebbing et al., 2019) that accounts for solid Earth deformation, perturbations in the gravitational potential, sea level changes due to rotational feedback from the GAL-induced mass variations, and global mass conservation. The Earth's deformation is assumed to be elastic, as is customary for sub-decadal phenomena.

We apply four different loads, comprising (a) atmospheric surface pressure variations, mass changes over (b) Greenland and (c) Antarctica, and (d) TWS anomalies over the remaining continents. The atmospheric loads are monthly mean pressures

from ERA-Interim, with values over the ocean replaced by their time-varying spatial mean; cf. Quinn et al. (2015). For terms (b)–(d), we use the satellite-based and leakage-corrected surface mass estimates described in Appendix B. In adopting these loads in the sea level equation, we assume that all continental mass changes are related to variations in water (or ice) mass and consequently added to or removed from the ocean. Fig. B1 illustrates the so-derived GAL fluctuations in space (bar the atmospheric contribution), while the total excitation signal ($\hat{\chi}^{\mathrm{L}}$) is shown in Fig. 1. On the interannual time scales considered

here, the atmospheric contribution to $\hat{\chi}^{\mathrm{L}}$ is very small, featuring a standard deviation of less than 0.20 mas.

A final, more subtle issue is that GAL effects may also arise from dynamic $p_{\mathrm{b}}$ variability, as driven by atmospheric forcing or intrinsic ocean processes, and not by land-ocean mass transfer (Vinogradova et al., 2015). These gravitational effects were omitted in the OCCIPUT simulations, so we account for them after the fact, in keeping with the ocean's tendency for an equilibrium response to body forces at periods longer than a few months. Restricting the loads to dynamic ocean mass changes

simplifies the GAL algorithm to a convolution of $p_{\mathrm{b}}$ anomalies with the proper Green's function (Vinogradova et al., 2015). We have applied a spherical harmonic version of this convolution (Schindelegger et al., 2018) to a few OCCIPUT ensemble members and the $p_{\mathrm{b}}$ ensemble mean. The derived excitation series were found to differ in detail but agree in terms of their interannual peaks (∼1 mas in $\chi_1$, ∼1.5 mas in $\chi_2$). Thus, for the purpose of this study, we represent the ocean-dynamics GAL signal using the $p_{\mathrm{b}}$ ensemble mean only and impose the implied mass redistribution on each of the 50 ensemble members. The

GAL effects due to ocean dynamics are therefore contained in $\hat{\chi}^{\mathrm{O}}$, and not in $\hat{\chi}^{\mathrm{L}}$.

*Author contributions.* **Lara Börger**: Data curation, Formal analysis, Investigation, Methodology, Software, Validation, Visualization, Writing - original draft, Writing - review & editing. **Michael Schindelegger**: Conceptualization, Formal analysis, Funding acquisition, Methodology, Supervision, Validation, Writing - original draft, Writing - review & editing. **Mengnan Zhao**: Data curation, Formal analysis, Writing - review & editing. **Rui M. Ponte**: Conceptualization, Funding acquisition, Writing - review & editing. **Anno Löcher**: Formal analysis,



Writing - review & editing. **Bernd Uebbing**: Formal analysis, Writing - review & editing. **Jean-Marc Molines**: Data curation, Writing - review & editing. **Thierry Penduff**: Funding acquisition, Writing - review & editing.

*Competing interests.* The authors declare no conflict of interests.

*Acknowledgements.* L.B. and M.S. were supported by the German Research Foundation (DFG, Project no. 459392861). Work at AER (R.P. and M.Z.) was supported by NASA through GRACE Follow-On Science Team grants 80NSSC20K0728 and 80NSSC24K1154. The work
from B.U. has been funded by KI-FOR Algorithmic Data Analytics for Geodesy (AlgoForGe) by DFG, project number 459420781. The results of this research have been achieved using the PRACE Research Infrastructure resource CURIE based in France at TGCC. This work is a contribution to the OCCIPUT and IMHOTEP projects. OCCIPUT has been funded by ANR through contract ANR-13-BS06-0007-01. IMHOTEP is being funded by CNES through the Ocean Surface Topography Science Team (OSTST).



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
