# Peer review of "Chaotic oceanic excitation of low-frequency polar motion variability"

_Earth System Dynamics, 2024_

## Author Comment (AC1)

**Comments by Referee #1:**
(Responses by the authors are highlighted in blue)

I read with interest the manuscript "Chaotic oceanic excitation of low-frequency polar motion variability" submitted by Börger et al. for possible publication in Earth System Dynamics. The paper utilizes the OCCIPUT large ensemble with 50 realizations of time-variable eddy-permitting ocean mass and flow fields to calculate effective ocean angular momentum functions characterizing the excitation of changes in the solid Earth's orientation with respect to inertial space. The paper is very well written und certainly fits into the scope of the journal. I recommend this work for publication as soon as a number of comments have been reasonably well addressed.

(1) The analysis presented in this paper is based on the SPACE2018 series of Earth Orientation Parameters as processed at the JPL. Authors should explain in more detail why SPACE2018 is used here instead of the associated COMB2018 series, or a more recent reprocessing of the same data (i.e., COMB2019). Authors should also consider to use the newly published EOP series from the ITRF2020 computation that are operationally updated as EOP 20 C04 by scientists from the Paris Observatory. In any case, it needs to be discussed in the article how a particular choice of the EOP series might affect the interpretations of the results presented here.

Use of SPACE2018 on our side has carried through from previous studies, but we will switch to newer solutions in future work. Repeats of our excitation budget analysis with the mentioned series (COMB2019/2022, EOP 20 C04) showed essentially no sensitivity to the choice of the rotation data: PVE values in Table 2 only changed by 0.2% at most. We will add this result as a footnote to Table 2 in the revised manuscript.

(2) Time-variable gravity field representations from GRACE that are additionally augmented by SLR and DORIS observations to extend the time-series have been used only rarely in Earth Orientation Parameter research. In view of the cautious comments provided by the authors in line 240, I propose to explicitly show the hydrological angular momentum functions derived from GRACE+SLR+DORIS for the whole time-period 1995-2015, and compare it with GRACE-based excitation functions -- ideally derived from publicly available Level-3 products, like the Cost-G combination solution available via gravis.gfz-potsdam.de to make results traceable -- and an independent model-based hydrologic excitation function published elsewhere. Please note that a detailed discussion of the contributions from Greenland and Antarctica is not necessary at this point.

Good idea, and we have in fact done such comparisons during our study. We would like to show **Figure S1** in a newly created supplementary, where we compare our low-pass filtered hydrological $\chi_{1,2}$ estimates with time series from the COST-G Level-3 product and the Global Land Data Assimilation System (GLDAS). Given that excitations from hydrological models are generally deemed uncertain (see the literature on the subject), we will refrain from mentioning the GLDAS results in the main text. However, we will point to some of the differences between the GRACE+SLR+DORIS solution and the COST-G time series, as these differences fit into the discussion near line 240 about possible remaining limitations in the utilized hydrological angular momentum estimates.

Note that in addition to Figure S1, our results have already been made traceable by providing all our (unfiltered) angular momentum time series on zenodo; see the "Data availability" statement.

[Figure]

**Figure S1**: Hydrological contribution to interannual polar motion excitation (mas) deduced from the GRACE/SLR/DORIS gravity field solution described in the main text (1995/01–2015/12, blue lines), the COST-G GravIS RL01 continental water storage anomalies (2002/04–2015/08 with gaps, yellow), and the GLDAS model (2002/01–2015/12, red). Each time series has been filtered to periods longer than 14 months, cut back by 4 months at the respective end points. Trends and mean of the COST-G and GLDAS time series have been adjusted such that they agree with the GRACE/SLR/DORIS trend and mean over a common time period starting early 2002. Note that GRACE/SLR/DORIS was detrended over 1995–2015, as in the main text.

(3) It is quite surprising to see that the largest interannual surface mass variations outside Greenland and Antarctica are found on the Malakka peninsula in South-East Asia. This is not really intuitive from a hydrometeorological perspective and calls for further investigations. In particular, it should be thoroughly checked if poorly treated tectonic signals associated with the 2004 Sumatra-Andaman earthquake (and later events in neighboring areas) are responsible for this feature. Please report in detail about any modifications made to the GRACE+SLR+DORIS processing, which is not yet really well covered in the scientific literature.

Thank you for this keen observation. Additional checks by us have shown that the large-magnitude surface mass variation near Phuket is indeed the manifestation of an uncorrected post-seismic signal. However, the anomaly is of very limited spatial extent and sits in low latitudes, such that the hydrological polar motion excitation remains virtually unaffected: When setting the terrestrial water storage fields over the area in question to 0, the PVE values in Table 2 changed by 0.2% in $\chi_1$ and by 0.4% in $\chi_2$ (brute force sensitivity test). For the revisions, we would like to point out the Phuket anomaly in the caption of Fig. B1 and add one sentence to Appendix B,

indicating that co- and post-seismic deformation signals have not been removed from the GRACE+SLR+DORIS solution.

(4) Authors speculate in both abstract and conclusions about possible implications of this work for EOP prediction, but fail to elaborate it further in the article. I suggest to remove this comment from the abstract in order to avoid raising unrealistic expectations with the reader. In any case, rigorously assessing the potential consequences for EOP predictions should be left for future study.

Thanks for the suggestion, but we disagree. It is quite natural and common in papers to highlight implications, even if they are not treated at length but nevertheless clear from the context – as is the case here. One could go even further and argue that such cross-links are part of the meat of interdisciplinary journals and papers. In any case, given the way how the EOP prediction aspect is mentioned in the last sentence of the abstract, there should be no ambiguity that it is meant as a future line of research.

---

## Author Comment (AC2)

**Comments by Referee #2:**
(Responses by the authors are highlighted in blue)

Börger et al. used the outputs of 50 ensemble OGCM simulations, driven by the same atmospheric data from the DRAKKAR forcing set (DFS 5.2) with slightly perturbed initial conditions. Then, the authors computed monthly OAM, separating the three output variables, $p_b$, and into the common signals to all 50 members and those uncommon signals (eq. 3). Based on those common and uncommon signals, the authors computed a series of OAM data and showed some interesting results. In particular, Figure 2c shows an intriguing peak in the "forced" signal (L169-170), which is clearly not annual but rather broad and significant; the same peak can also be found in Figure 3c; I have never seen such a peak that is unexpected and deserves to be analyzed in more detail. This broad spectral peak between periods of ~1.2 and ~2.5 years, apparent particularly in the mass term (Figure 2c in the main text) is indeed interesting. It is not a peculiarity of the OCCIPUT OAM data, as analyzed here, but a common feature across many ocean models; see **Figure S2**. We are currently writing up results of another study that attributes parts of this interannual OAM signals to specific patterns of ocean mass change. However, this is clearly another paper and does not fit the scope of the present work.

[Figure]

**Figure S2**: Smoothed amplitude spectra of the $\chi_2$ component of oceanic polar motion excitation (mas), separated into (a) mass and (b) motion terms. Shown are versions from four different models, comprising the OCCIPUT (forced component only), a modern ocean state estimate (ECCOv4 release 4b), MPIOM by GeoForschungZentrum Potsdam (i.e., the series most frequently used in Earth rotation studies), and ORAS5 (as analyzed by Börger et al. 2023).

The goal of this paper is to "separate the ensemble OAM estimates into forced and intrinsic components and assess their contribution to the observed wobble excitation" (L44-45). However, I wonder if the goal can be accomplished from the presented approach. While the authors consider the common signals as "forced" and the uncommon signals as "intrinsic" (or "chaotic" in places), I do not agree with the authors' understanding (or their terminology). I understand that the truly "intrinsic" signals should also be included in the "forced" ones and that the uncommon signals after the ensemble simulations are simply due to different initial and boundary conditions, indicating simply the uncertainties of the OGCM simulations, whereas it is important to quantify the uncertainties. Thus, instead of agreeing with a statement in Line 279 "*variability in $\chi^{o,i}$ is a considerable fraction (43–50%, see Sect. 3.1) of the total oceanic excitation of polar*

*motion even on interannual time scales*", I have rather understood that the simulation outputs have considerable uncertainties. I would suggest changing the scope of this work and focusing more on the analysis of the derived interesting "forced" OAM signals.

In relation to the reviewer's main point, we wish to clarify the following:

- **We separate forced and intrinsic variabilities from an ensemble simulation with perturbed initial conditions and same atmospheric forcing, which is a standard and very robust modelling approach in geosciences**. Besides a huge number of atmosphere and climate studies (see, e.g., Nikiéma & Laprise 2016 and Maher et al. 2020, respectively, and references therein), many oceanographic papers have adopted this approach using various models at different resolutions over several regions (e.g., Combes and di Lorenzo 2007, Hirschi et al. 2013, Gehlen et al. 2020, Uchida et al. 2021, Leroux et al. 2022, Benincasa et al. 2024), as well as the 19 OCCIPUT papers published since 2014 (https://meom-group.github.io/projects/occiput/) including those cited in our manuscript.

- The 50 OCCIPUT ensemble members are identical in terms of the underlying model, parameterizations, boundary conditions, and atmospheric forcing. All members share the same uncertainties in these components and whatever impact they may have on OAM: **The ensemble spread does not come from model errors. Instead, it comes from inherently non-linear ocean dynamics**. The spread is triggered by a weak stochastic perturbation that is temporarily applied to the density equation following a common spin-up (cf. lines 110–113, Section 4.2 in Bessières et al. 2017). Growth and saturation of the spread occurs quickly in turbulent, eddy-rich areas (Penduff et al. 2014, Bessières et al. 2017) and more slowly in less unstable regions, as predicted by instability theory. Please check out Figure 2 in Penduff et al. (2014) for a compelling illustration of this concept, and the role of mesoscale instabilities in the origin of the ensemble spread.

The substantial and large-scale inter-member differences that are here diagnosed from the saturated ensemble spread are thus only due to the non-linearly induced random phase of intrinsic variability within each member, whose dispersion is triggered by initial perturbations. This standard approach in estimating forced and intrinsic variabilities in the ocean and other components of the climate system is robust and has led to many results in the literature. The work presented here specifically highlights the impact of non-linear ocean dynamics on a globally-integrated quantity relevant to geodesy and solid Earth research.

The problem might be because the authors used "eddy-permitting" model instead of "eddy-resolving" model. If the latter "eddy-resolving" model was used, it would have much finer spatial resolution and allows to more accurately compute the fine-scale ocean dynamics; the "atmospheric-driven" component should also more accurately include the intrinsic chaotic ocean variability. The authors might be recognizing this point in view of the sentence in Lines 279-280, "*A caveat to be acknowledged…*".

Some authors have indeed suggested that fine horizontal resolution may be beneficial for modeling wind-driven OAM changes on intraseasonal time-scales (Afroosa et al. 2021, Harker et

al. 2021, Afroosa et al. 2022); see also line 286. However, these benefits are not as clear-cut as your remark implies, and we do not know of any study that has looked into this question on interannual time scales.

More importantly, and for the purpose of the present work, the use of an eddy-permitting (here 1/4°) model instead of an eddy-resolving (e.g., 1/12°) model is not a major limitation: 1/4° and 1/12° ocean model simulations have been shown to be consistent regarding the existence, origin, spatial structure, and spatio-temporal scales of intrinsic variability (Sérazin et al. 2015, Gregorio et al. 2015). The magnitude of intrinsic variability is a bit higher in the 1/12° simulations for certain variables, but only barely so on interannual time scales (see Figure 9 in Sérazin et al. 2015 for the case of sea level). For other variables however, interannual intrinsic variability amplitudes at 1/4° and 1/12° are barely distinguishable (see Figure 7 in Gregorio et al. 2015 for the case of AMOC). Hence, the OCCIPUT large ensemble at eddy-permitting resolution is a good choice for venturing a first look into the effects of oceanic chaos on OAM. Also note that performing the 1/4° OCCIPUT ensemble required 20 million CPU hours: performing the same exercise at 1/12° resolution would cost about $3^3$ times more (i.e., about 600 million CPU hours), which lies far beyond the computing power presently available to research teams.

**Minor comments:**

Line 56: There are still large uncertainties in $Q_c$, and the value 179 is rather high.
$Q_c$ = 179 is a standard choice that has been used countless times before. We have rerun our excitation budget analysis with much lower values (e.g., $Q_c$ = 50, as advocated for by Yamaguchi and Furuya, 2024) and found negligible impacts on the results: PVE values given Table 2 changed by 0.0 to 0.1%, and by 0.3% in only one case.

Line 248: Is there any evidence for the effect of the core on interannual wobble excitation? While the present work assumes pure elastic deformation, 1.10 and 1.608 in A1, anelastic deformation will rather need to be considered in longer timescales.
Chen et al. (2019), building on work by Kuang et al. (2019), presented tentative evidence for core effects on interannual wobble excitation, particularly near the 6-year period. One of our arguments is that such inferences can be complicated by errors in the corrections for surficial mass redistributions and particularly intrinsic OAM signals; see lines 255–261.

As for the impact of anelastic deformation on the $\chi$ functions, Wahr (2005) showed that this would change the real-valued scaling factors (two-digit version of Eq. A1) from 1.10 and 1.61 to (1.10 − i·0.01) and (1.61 − i·0.02), respectively. These changes amount to ≤ 1.5%, comparable to the uncertainty of other numerical constants and assumptions in the excitation formalism (Gross 2007). It is therefore not unjustified to neglect these small imaginary components, as done here. We will add a brief note on this matter just above Eq. (A2) in the revised manuscript.

**References**: (asterisk marks papers not cited in the main text)

Afroosa et al. (2022): https://link.springer.com/article/10.1007/s10236-022-01518-8 [*]
Benincasa et al. (2024): https://doi.org/10.5194/os-20-1003-2024 [*]
Bressières et al. (2017): https://doi.org/10.5194/gmd-10-1091-2017
Chen et al. (2019): https://doi.org/10.1029/2019JB018541
Combes and di Lorenzo (2007): https://doi.org/10.1016/j.pocean.2007.08.011 [*]
Gehlen et al. (2020): https://doi.org/10.1029/2020GL088304 [*]
Gregorio et al. (2015): https://doi.org/10.1175/JPO-D-14-0163.1 [*]
Gross (2007): https://doi.org/10.1016/B978-044452748-6.00057-2
Hirschi et al. (2013): https://doi.org/10.5194/os-9-805-2013 [*]
Kuang et al. (2019): https://doi.org/10.1016/j.geog.2019.06.003
Leroux et al. (2022): https://doi.org/10.5194/os-18-1619-2022 [*]
Maher et al. (2020): https://doi.org/10.5194/esd-12-401-2021 [*]
Nikiéma & Laprise (2016): https://doi.org/10.1007/s00382-015-2604-3 [*]
Penduff et al. (2014):
https://www.researchgate.net/publication/283497595_Ensembles_of_eddying_ocean_simulati
ons_for_climate
Sérazin et al. (2015): https://doi.org/10.1175/JCLI-D-14-00554.1 [*]
Uchida et al (2021): https://doi.org/10.3390/fluids 6060206 [*]
Wahr (2005): see manuscript bibliography
Yamaguchi and Furuya (2024): https://doi.org/10.1186/s40623-023-01944-y [*]

---

## Author Comment (AC3)

**Comments by Referee #3** (Christian Bizouard):
(Responses by the authors are highlighted in blue; note that the line breaks in the original comment have been changed for clarity)

The main interest of that paper is to show that the "intrinsic" circulation in the ocean excites significantly the polar motion at both inter-seasonal and interannual time scales. Stemming from meso-scales eddies, this intrinsic circulation is chaotic, and till now is not fully captured by oceanographic observations. However, it can be simulated, as done by the authors. In many respects, the authors are reshuffling the deck when it comes to modelling polar motion. They show that the inter-annual polar motion could result from the oceanic circulation, without the need of the core-mantle interaction as advocated by many recent papers.
Thank you for assessing and appreciating our work, Christian.

I am not sure whether the authors realized that their study could also modify the current understanding of the Chandler wobble. Indeed, according to Fig. 2b, the mass term of the intrinsic excitation at the level of 1 mas could contribute to Chandler wobble in a very significant way.
Yes, this implication was and is well on our mind. We have quite compelling results on the Chandler wobble excitation by intrinsic oceanic signals but decided not to squeeze them into the current manuscript but reserve them for another paper. The relatively short analysis window (1995–2015) is a bit of an issue, though.

In the forced part of the ocean angular momentum, I wonder whether the authors considered the pole tide as a source of forcing. Could the authors address that important question in the revised version?
Any pole tide signal, which at interannual periods is dominated by an oscillation at the Chandler frequency, is removed from the observations by means of the deconvolution operator (Eq. 1). As a static and thus dynamically irrelevant phenomenon, the pole tide is also omitted from OCCIPUT, just as in any other ocean model. Hence, the treatment of the pole tide in our study is consistent among the geodetic and oceanic excitation, i.e., it is absent from both series. We will insert a corresponding note in the revised manuscript (e.g., in Sect. 2.2 "Oceanic excitation").

The paper is well written, the approach is well presented. Only legends in Fig. 5 and 6 and captions of tables 1/2 deserve some light improvements: the legends for excitation functions have to be well split, in tables write "Percentage of Explained Variation (PVE)".
Thank you for the hints, we will consider them in the revised manuscript.